# Genetic and Pathogenic Characterization of QX(GI-19)-Recombinant Infectious Bronchitis Viruses in South Korea

**DOI:** 10.3390/v13061163

**Published:** 2021-06-17

**Authors:** So-Youn Youn, Ji-Youn Lee, You-Chan Bae, Yong-Kuk Kwon, Hye-Ryoung Kim

**Affiliations:** Avian Disease Division, Animal and Plant Quarantine Agency, Gimcheon 39660, Korea; syyoun@korea.kr (S.-Y.Y.); enteric@korea.kr (J.-Y.L.); kyusfather@korea.kr (Y.-C.B.); kwonyk66@korea.kr (Y.-K.K.)

**Keywords:** infectious bronchitis virus, recombinant, chicken, genotype GI-19

## Abstract

Infectious bronchitis viruses (IBVs) are evolving continuously via genetic drift and genetic recombination, making disease prevention and control difficult. In this study, we undertook genetic and pathogenic characterization of recombinant IBVs isolated from chickens in South Korea between 2003 and 2019. Phylogenetic analysis showed that 46 IBV isolates belonged to GI-19, which includes nephropathogenic IBVs. Ten isolates formed a new cluster, the genomic sequences of which were different from those of reference sequences. Recombination events in the *S1* gene were identified, with putative parental strains identified as QX-like, KM91-like, and GI-15. Recombination detection methods identified three patterns (rGI-19-I, rGI-19-II, and rGI-19-III). To better understand the pathogenicity of recombinant IBVs, we compared the pathogenicity of GI-19 with that of the rGI-19s. The results suggest that rGI-19s may be more likely to cause trachea infections than GI-19, whereas rGI-19s were less pathogenic in the kidney. Additionally, the pathogenicity of rGI-19s varied according to the genotype of the major parent. These results indicate that genetic recombination between heterologous strains belonging to different genotypes has occurred, resulting in the emergence of new recombinant IBVs in South Korea.

## 1. Introduction

Infectious bronchitis (IB) is a highly contagious disease of poultry caused by the infectious bronchitis virus (IBV); the disease can lead to significant economic damage to the poultry industry [1,2]. IBV, which affects chickens of all ages and species, causes respiratory, urinary, and reproductive infections [3]. Furthermore, damage to the tracheal cilia means that chickens infected with IBV commonly develop secondary infections caused by bacteria or other pathogens; these secondary infections have a higher mortality rate [4].

The IBV (family Coronaviridae) genome encodes spike (S), envelope (E), membrane (M), and nucleocapsid (N) structural proteins. The *S* gene is used for the molecular characterization of IBV isolates. The S protein, especially the S1 subunit (which is highly variable among IBVs and harbors antigenic epitopes), is associated with antigenic neutralization, hemagglutination, immune protection, and cell tropism [5,6,7]. Therefore, variations in the S1 subunit of the S protein mainly determine the genotype, antigenicity, and pathogenicity among IBVs [8,9,10].

Many antigenic variants, serotypes, and field strains of IBV have been isolated [11]. At least 30 different IBV serotypes have been identified worldwide, and most available IBV vaccines do not provide complete cross-protection against viruses with different serotypes [12]. IBV variants are a major problem for the poultry industry. Outbreaks caused by recombinants between GI-19 and GI-13 genotypes, which emerged in China in 2016, have become common in recent years [13]. Additionally, recombinant TW IBVs between GI-19 and GI-7 have begun to spread [11]. Recombinants arise due to the high error rate of the viral RNA-dependent RNA polymerase, which results in a high frequency of genetic mutations (e.g., gene insertions, mutations, deletions, and reconstructions) during RNA replication [6]. Owing to the continuous appearance of IBV variants showing a possible shift in serotype or pathogenicity, outbreaks of IB occur frequently, even among vaccinated flocks [14].

Several indigenous and common IBV genotypes, such as the GI-15, GI-19 subgroups (KM91-like), and the GI-19 subgroup (GI-19-like), co-circulate in South Korea, and recombination events are thought to have occurred. The GI-15 and KM91-like IBVs, which are native viruses associated with localized outbreaks, have been co-circulating since the 1990s [15]. In particular, the KM91-like IBV (which is nephropathogenic) is associated with a high mortality rate and has caused great economic losses to the poultry industry, despite the availability of a vaccine. QX-like IBVs were introduced into South Korea between 2002 and 2003 [15,16,17]. Moreover, variant IBVs (recombinants between KM91-like and QX-like viruses) emerged in South Korea in 2005 and have been isolated continuously since then [18]. IB outbreaks are common, and are associated with high mortality due to nephritis and respiratory disease, despite repeated vaccinations. Therefore, we conducted genetic and pathogenic analyses of Korean IBVs isolated from 2003 to 2019 to find out whether another IBV recombination event has occurred, and whether recombinant IBVs show altered pathogenicity.

## 2. Materials and Methods

### 2.1. Viruses

Fifty-six IBV isolates were collected from infected chickens in South Korea between 2003 and 2019 (Table 1). The IBV isolates (40 from broiler, five from layer, one from breeding, and ten from native chickens) were obtained from five provinces (ten from Gyeongsang, 15 from Jeolla, 17 from Chungcheong, two from Gangwon, and 12 from Geyonggi). As Gangwon has fewer chickens than the other provinces, the number of isolates obtained from that province was less than that from other provinces.

### 2.2. Viral RNA Extraction, RT-PCR, and Sequencing

Ten-day-old specific-pathogen free (SPF) embryonated chicken eggs were used for isolation and propagation of IBVs. Allantoic fluid from eggs infected with each isolate was harvested after incubation at 37 °C for 72 h, and then frozen at −70 °C until use. Viral RNA was extracted from virus-infected allantoic fluid using a QIAamp viral RNA mini-kit (Qiagen, Hilden, Germany). The complete *S* gene sequence was reverse transcribed using a Super-Script III reverse transcriptase kit (Invitrogen, Carlsbad, CA, USA) and AccuPrime Taq DNA Polymerase High Fidelity (Invitrogen). Amplification of the *S* gene was performed using specific primers, as described previously [19]. Gene sequencing was carried out using the custom sequencing service provided by Bionics Co., Ltd. (Daejeon, Korea). The nucleotide sequences for the *S* genes have been submitted to GenBank under accession numbers MW984619–MW984674.

### 2.3. Phylogenetic Analysis, Sequence Comparisons, and Recombination Analysis

BioEdit software version 7.0.9.0 was used to analyze and edit the generated nucleotide sequences of the *S1* gene from IBV isolates [20]. The IBV reference strains (four genotypes: G1-15, KM91-like, QX-like, and GI-1) were imported from the GenBank database. Sequence analyses and alignment of the *S1* gene were performed using Clustal W. The phylogenetic tree was constricted in MEGA, version 6, using the neighbor joining method, with 1000 bootstrap replicates [21].

To identify recombination events, the *S1* genomic sequence was compared with those of QX-like, GI-15, and KM91-like genotype strains (QXIBV, LX4, K210-01, and KM91). Consecutive IBV nucleotide sequences from the *S1* gene, based on the multiple alignment results, were used for similarity plotting analysis using the Simplot program (v 3.5.1), with a window size of 200 bp and a step size of 20 bp [22].

### 2.4. Pathogenicity Testing

All animal experiments were approved and supervised by the Institutional Animal Care and Use Committee (IACUC) of the Animal and Plant Quarantine Agency (APQA) of South Korea (permission number 2019-197). One-week-old white leghorn SPF chickens were purchased from a local company (Namduk SPF, Gyeonggi, the South Korea) and divided randomly into four groups (*n* = 20/group). The birds were housed in separate isolation units (Three-Shine INC., Daejeon, the South Korea). Three groups were infected with 100 μL of IBV (10^6.5^ EID_50_ per 0.1-mL dose) via the oculonasal route, and one group was infected with a phosphate-buffered saline solution. Clinical signs (sneezing, tracheal rales, and chills) were monitored daily and recorded for 21 days post-infection (dpi). Oropharyngeal (OP) and cloacal (CL) swabs were collected at 3, 5, 7, 10, 14, and 21 dpi. Three (at 3 and 7 dpi) and four (at 14 dpi) birds per group were selected randomly and sacrificed humanely. The remaining ten birds in each group were euthanatized at 21 dpi. Trachea, kidney, and cecal tonsil (CT) samples were collected carefully. To measure tracheal ciliostasis, tracheal rings were cut from the dissected trachea and examined under an inverted microscope (Eclipse Ts2, Nikon, Tokyo, Japan). The degree of ciliostasis was scored as follows: 0, 100% of cilia beating; 1, 75% of cilia beating; 2, 50% of cilia beating; 3, 25% of cilia beating; and 4, no cilia beating [23].

IBV replication in the trachea, kidney, and CT, as well as shedding in OP and CL swabs, were examined using IBV-specific primers (IBV5′GU391 and IBV5′GL533) and a probe (IBV5′G) [24]. Viral RNA was extracted from samples and swabs using a QIAamp viral RNA mini-kit (Qiagen, Hilden, Germany). Quantitative RT-PCR was performed using a Multiplex RNA Virus Master mix (Roche Diagnostics, France). The same titrated stock of virus was used to extract and dilute each viral RNA, which was used to establish the standard curve for each viral RNA to challenge the chickens. The results indicate EID50/mL. The assay for the detection was limited to 10^1.6^–10^1.9^ EID_50_/mL.

### 2.5. Statistical Analysis

The statistical analysis of virus replication from the animal trials was performed by means of ANOVA with Bonferroni’s multiple comparison test. The following notations are used to indicate significant differences between groups: * *p* < 0.05; ** *p* < 0.01; *** *p* < 0.001.

## 3. Results

### 3.1. Phylogenetic Analysis

The IBV isolates clustered into two different genetic groups (GI-15 and GI-19). We divided GI-19 into two subgroups (KM91-like and QX-like). The S1 gene showed that the 46 isolates were classified as QX-like (40 isolates) or KM91-like (six isolates), whereas ten were classified as a new cluster, distinct from GI-19, GI-15, and GI-1 (Figure 1).

### 3.2. Sequence Comparisons

The sequencing results showed that the *S1* genes of the IBV isolates contained insertions or deletions, resulting in a different number of nucleotides (1584–1596); hence, the *S1* gene of the 56 IBV isolates encoded between 528 and 532 amino acids (data not shown).

Most IBVs circulating in China belong to the QX-like genotype and can be classified into two clusters [25]. Therefore, two strains, QXIBV and LX4, were used as parental strains to compare IBV sequences. The nucleotide and amino acid sequences of each virus were compared with those of the QXIBV (QX-like cluster I), LX4 (QX-like cluster II), KM91 (KM91-like), and K210-01 (G1-15) strains. The *S1* genes of 40 QX-like IBV isolates showed high nucleotide identity with QXIBV (94.8–97.5%) and LX4 (94.4–96.3%). However, rGI-19-I (92% and 90%) rGI-19-II (89–90% and 88%), and rGI-19-III (79% and 79%) showed relatively low identity with the 40 QX-like IBV isolates. Additionally, the identity between KM91 and K281-01 with rGI-19-I (91% and 79%), rGI-19-II (93% and 79%), and rGI-19-III (80% and 89%) was variable (Figure 2).

### 3.3. Recombination Analysis

The Simplot program (version 3.5.1) is able to prove the presence of potential recombination regions based on the *S1* genes. The QXIBV, LX4, KM91, and K281/01 strains were employed as putative parental strains, and the 56 IBV isolates were taken as queries in the Simplot analysis. Strains were fixed as recombinants if there is occurrence of any crossover between the putative parental strains. The analysis revealed three patterns based on the crossover regions (Figure 3). Nine IBV recombinants (rGI-19-I and rGI-19-II) between QXIBV and KM91, and one IBV recombinant (rGI-19-III) between QXIBV and K281/01 were analyzed. The major parental strain of rGI-19-I was QXIBV, and the partial *S* gene might have been acquired from KM91. In addition, rGI-19-II was a recombinant of QXIBV and KM91. However, the insert position and the length of the partial *S* gene differed from those in rGI-19-I. The length of QXIBV (a parent strain of rGI-19-II) was longer than that of rGI-19-I. The major parent strain of rGI-19-III was K281/01, and the partial *S* gene might have been acquired from QXIBV; the *S1* gene contained two potential crossover sites (724 and 1102 bp), and three different characteristic nucleotide regions, similar to those in K281/01 and QXIBV. Two regions of the *S1* gene sequence showed high similarity with K281/01 (90% and 94% at the nucleotide (nt) level and 85% and 95% at the amino acid level) and the other part of that with QXIBV (nt 96% and aa 92%). These analyses provide important evidence that recombinant IBV isolates (rGI-19-I, rGI-19-II, and rGI-19-III) descended from three putative parents: KM91-like, QX-like, and GI-15. The results of recombination analysis demonstrate that heterologous recombination occurred between IBV isolates.

### 3.4. Animal Experiments and Histopathology

In recent years, two kinds of recombinant (rGI-19-II and rGI-19-III) have emerged in South Korea. Therefore, to compare the pathogenicity of recombinants rGI-19-II and rGI-19-III with that of a non-recombinant (GI-19), we inoculated them into 1-week-old SPF chickens via the oculonasal route. A 7 or 14 dpi, we found that the tracheal ciliostasis of the three infected groups was almost completely inhibited compared with that in the mock-infected group. Of note, tracheal ciliostasis in chickens treated with rGI-19-II and GI-19 recovered almost completely by 21 dpi, whereas this was not the case for chickens receiving rGI-19-III (Table 2).

Histopathology revealed that rGI-19-III had caused tracheal lesions until 21 dpi. The tracheal lesions involved the loss of both cilia and epithelial cells, epithelial cell degeneration, epithelial cell hyperplasia, and infiltration of the surface and lamina propria layers by inflammatory cells (Figure 4). None of the challenge or mock-infected groups exhibited lesions in the other tissues.

The tracheas of chickens infected with GI-19 displayed no evidence of infection at 3 dpi; however, infection by rGI-19s was detected. At the same point, the titers of rGI-19-III were not the highest among other infected groups in the tracheas by 14 dpi, but the titer of rGI-19-III was higher than those of G1-19 at 21 dpi (*p* < 0.05). Overall, tracheal infection by rGI-19s was detected earlier, and with higher titers, than infections by GI-19 (Figure 5A). The viral load in kidney samples from chickens infected with rGI-19-III was considerably lower than that in the other groups at 7 and 14 dpi (*p* < 0.01). Although the titers of GI-19 increased gradually after infection, peaking at 7 dpi before falling slightly, those of rGI-19-II remained constant up to 14 dpi, and those of rGI-19-III increased markedly in 21 dpi (*p* < 0.05). rGI-19 titers tended to increase at a slower rate than those of GI-19 (Figure 5B). In CT, the titer of viruses increased after infection. However, the titers of rGI-19 were higher than those of GI-19 at 3, 7, and 14 dpi (*p* < 0.05, *p* < 0.01, and *p* < 0.001) (Figure 5C). Thus, we speculate that rGI-19s may be better at infecting the trachea than GI-19, but less able to infect the kidney (Figure 5).

At 3, 5, 7, 10, 14, and 21 dpi, infected and mock-infected groups shed virus through the OP and CL routes. Viral shedding by all infected groups was detected up to 21 dpi. The titers of rGI-19s shed via the CL routes were lower than that of GI-19 at 3, 5, 7, 21 dpi (*p* < 0.05, *p* < 0.01, and *p* < 0.001). Additionally, the amount of virus shed via the OP route decreased from 7 dpi (GI-19 and rGI-19-II) and 5 dpi (rGI-19-III). Despite these differences in titer, all viruses were shed via the OP and CL routes by 21 dpi, with GI-19 and rGI-19s showing similar patterns (Figure 6).

## 4. Discussion

Previous studies of IBV show that crossover sites occur within relatively conserved sequences close to the hypervariable region (HVR) of the *S1* gene, in a conserved sequence within the HVR, and in the *S2* gene [26,27,28]. Among the 56 Korean IBV isolates examined herein, 46 were classified as QX-like (40 isolates) or KM-91-like (six isolates); recombination events were detected in the remaining 10 isolates. All of the recombinant IBV isolates harbored crossover events in the *S1* gene. We performed a detailed examination of three recombinant types: recombinants rGI-19-I and rGI-19-II (resulting from recombination between the GI-19 and KM91-like genotypes) and recombinant rGI-19-III (resulting from recombination between the GI-19 and GI-15 genotypes). From 2005 to 2019, rG1-19-II was detected consistently in different regions of South Korea. Therefore, recombinant IBVs have the potential to spread widely across poultry farms in South Korea (Table 1 and Figure 3). The data also indicate that this site (625 bp) may serve as a major region for template switching during viral RNA synthesis, and as a breakpoint candidate for genetic recombination within the *S* gene of nephropathogenic IBV isolates in South Korea. Genetic recombination can occur within multiple genes of IBVs [29,30,31], which is a fact that should be considered in future research.

The emergence of recombinant variants of IBV, caused by genetic mutation, has been reported in many countries. Indeed, the use of live vaccines could lead to the emergence of new variant viruses via the recombination of field strains with vaccine strains [32]. Three kinds of live vaccine have been used in South Korea (a GI-1 genotype vaccine in 1986, a KM91-like genotype vaccine in 2009, and a QX-like genotype vaccine in 2018) [17]. The IBV isolates examined in the present study were obtained both before and after the introduction of these live-attenuated IBV vaccines. Our data show evidence of recombination between IBVs isolated between 2003 and 2019 (Figure 1, Figure 2 and Figure 3). Therefore, it is impossible to know whether the new IBV isolates identified in this study were generated naturally through the recombination of GI-15, KM91-like, and QX-like field strains.

The *S* proteins of different genotypes of IBV have been identified as being associated with the emergence of novel variants with different tissue tropisms [5,14,33,34]. Therefore, it is necessary to investigate potential differences in pathogenicity between recombinant IBV isolates. IBV isolates exhibit various tissue tropisms (e.g., respiratory, nephrotic, and gastrointestinal). In particular, the GI-19 strain shows stronger tropism and higher pathogenicity in the kidney than in the respiratory system. The GI-19 genotype also exhibits renal tropism, causing gross lesions that are pale, as well as swollen kidneys, with distention of the ureter and excess production of urates [25,35]. However, none of the recombinant isolates in the present study caused severe clinical signs or pathologic lesions, especially nephritis, even though they harbored the partial *S* gene of GI-19 genotype strains and were detected in the trachea, kidney, and CT. Therefore, the results indicate that despite recombination with a GI-19 type IBV, the recombinant viruses are less likely to cause nephrotic syndrome in chickens.

The cilia in the trachea play an important role in innate immunity, and help to prevent infection by pathogenic microorganisms [36]. GI-19 and rGI-19s caused ciliostasis and lesions in the trachea at 7 dpi and 14 dpi. In particular, rGI-19-III, which has GI-15 as its major parent, generated more severe lesions than the other viruses. The combination of lesions and ciliostasis may decrease immunity and increase susceptibility to secondary infections. Thus, rGI-19s may be associated with a greater risk of secondary infection than GI-19.

Since the cross-protection provided by IB vaccines is limited, the failure of vaccine-induced immunity has been reported, as recombinants continue to emerge [37]. Therefore, it is crucial to gain a better understanding of the antigenicity and pathogenicity of recombinant GI-19 IBVs. Our data suggest that recombination between co-circulating nephropathogenic and respiratory IBVs is occurring in South Korea. Moreover, it is possible that these events might increase the genetic diversity of IBVs in the field, thereby preventing effective disease control. The efficacy of current vaccines against these isolates should be the subject of further study. Continuous testing of the pathogenicity and serotype of new isolates remains crucial to improving the epidemiological understanding and control of IB.

## Figures and Tables

**Figure 1 viruses-13-01163-f001:**
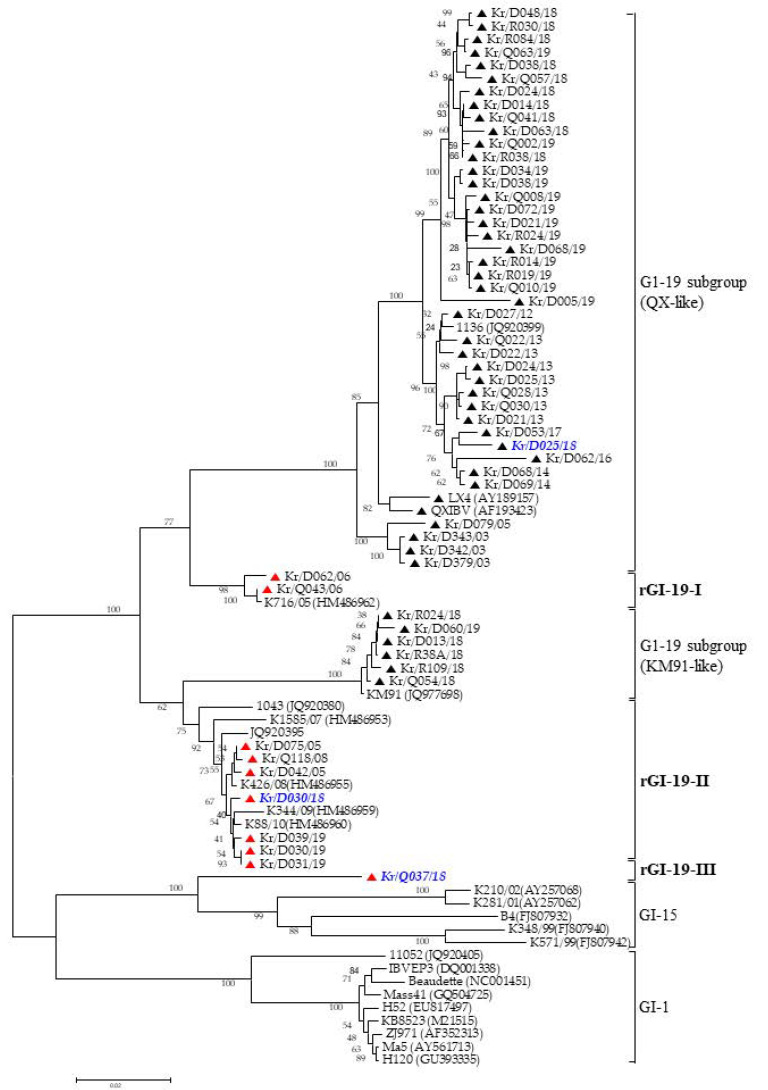
Phylogenetic tree of the *S1* nucleotide sequences of 56 IBVs. The recombinant IBVs in this study are marked with a red solid triangle. Other IBVs are marked with a black solid triangle. IBVs are used to pathogenicity test are marked with blue and italics. The provisional designations, including genogroups and sub-genogroups, are indicated on the right.

**Figure 2 viruses-13-01163-f002:**
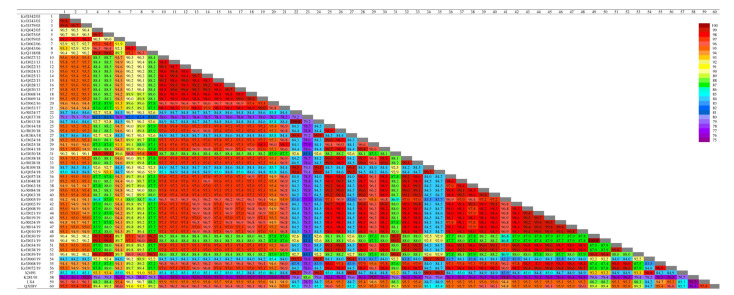
Percentage identity matrix of the nucleotide sequences of 56 isolates and 4 reference strains of IBV. The color-coded pairwise identity matrix was generated from the *S1* genes of 60 IBVs. Each colored cell represents the percentage identity score between two sequences (indicated horizontally to the right). The color key indicates the degree of correspondence between pairwise identities and the colors displayed in the matrix.

**Figure 3 viruses-13-01163-f003:**
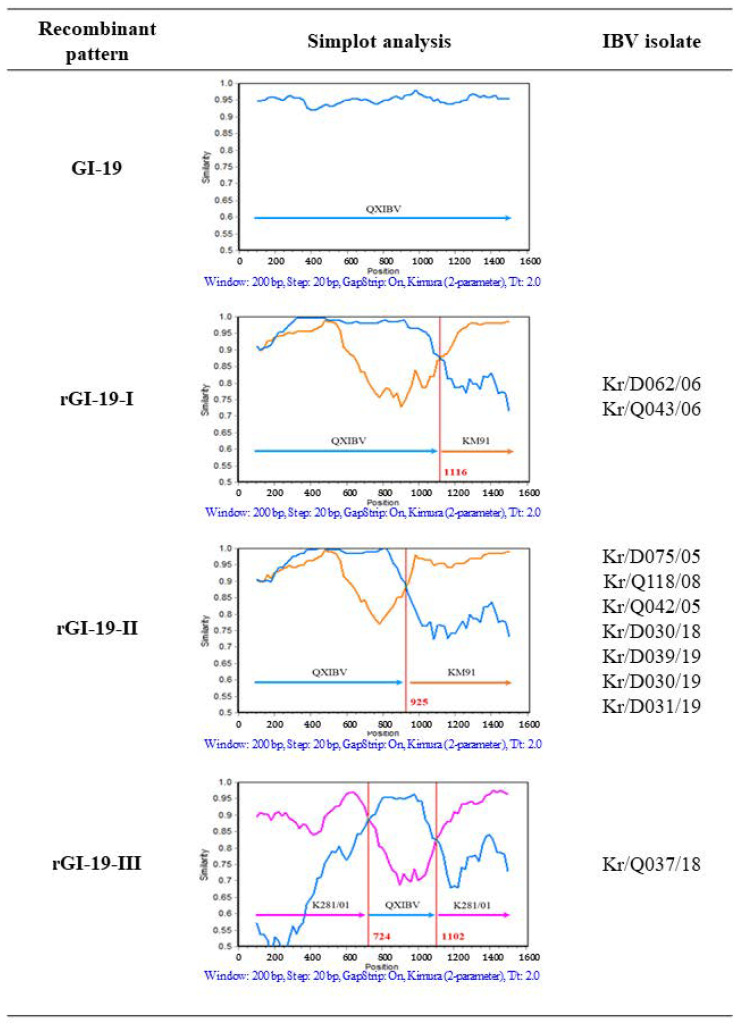
Recombination analysis in the *S1* gene of ten IBV isolates and their putative parents (QXIBV (blue), KM91 (orange), and K281/01 (pink)). The *y*-axis shows the percentage identity within a sliding window that is 200 bp wide and centered on the position plotted, with a step size between plots of 20 bp. The red vertical line denotes the recombination point.

**Figure 4 viruses-13-01163-f004:**
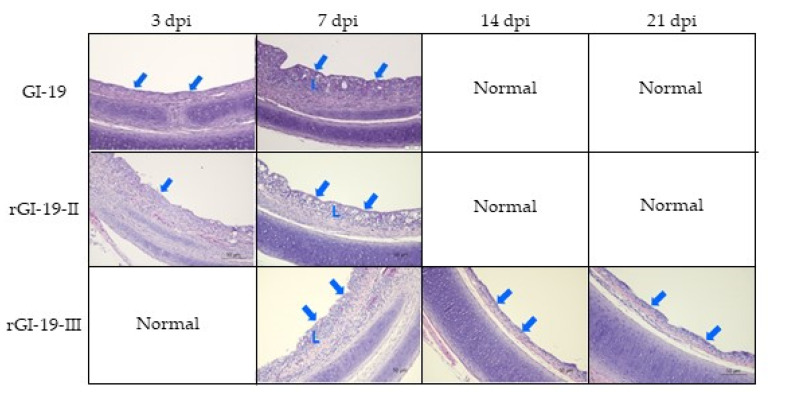
Histopathologic analysis of tracheas from birds infected with GI-19 or rGI-19s. Loss of cilia and cuboidalization of the surface epithelium denoted by arrows, inflammatory cells infiltration in lamina propia by L.

**Figure 5 viruses-13-01163-f005:**
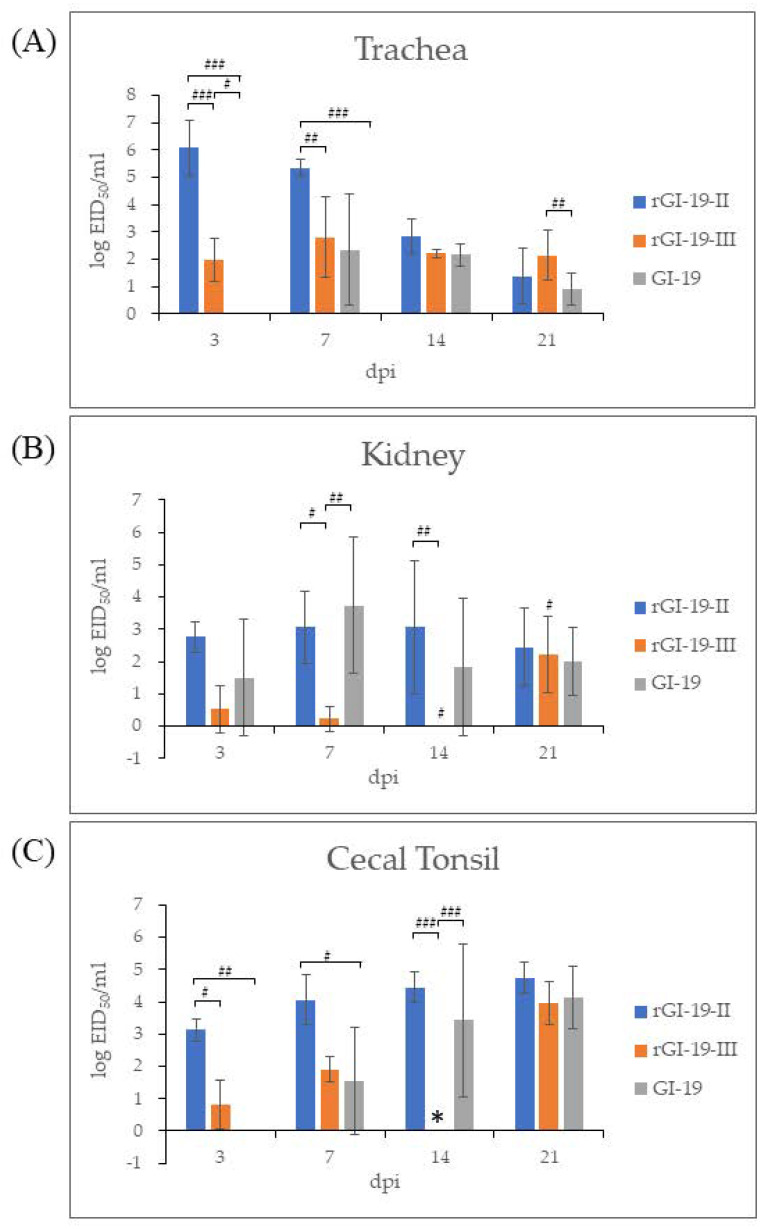
IBV viral load in different tissues (trachea (**A**), kidney (**B**), and cecal tonsil (**C**)) from chickens, as measured by RT-qPCR. The error bars represent the standard deviation. *: Negative result may be due to errors during sample processing. #: *p* < 0.05; ##: *p* < 0.01; ###: *p* < 0.001.

**Figure 6 viruses-13-01163-f006:**
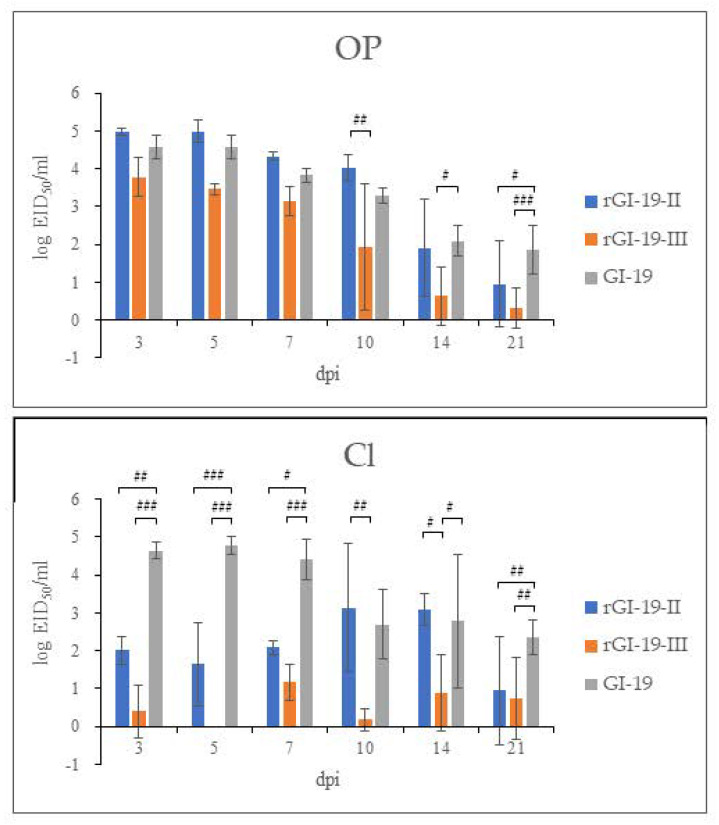
Mean viral shedding of infected IBVs, as measured by RT-qPCR. Each data point represents the IBV titer detected in oropharyngeal (OP) and cloacal (CL) swabs at different days post-infection (dpi). Bars represent the standard deviation of the mean. #: *p* < 0.05; ##: *p* < 0.01; ###: *p* < 0.001.

**Table 1 viruses-13-01163-t001:** Korean IBV isolates analyzed in this study.

IBV Strain	Year of Isolation	Type of Chicken	Location	Accession Number
Kr/D342/03	2003	Broiler	Chungcheong	MW984619
Kr/D343/03	2003	Breeding	Chungcheong	MW984620
Kr/D379/03	2003	Broiler	Chungcheong	MW984621
Kr/Q042/05	2005	Broiler	Jeolla	MW984622
Kr/D075/05	2005	Broiler	Chungcheong	MW984623
Kr/D079/05	2005	Native	Geyonggi	MW984624
Kr/D062/06	2006	Broiler	Gangwon	MW984625
Kr/Q043/06	2006	Broiler	Geyonggi	MW984626
Kr/Q118/08	2008	Broiler	Chungcheong	MW984627
Kr/D027/12	2012	Broiler	Chungcheong	MW984628
Kr/D021/13	2013	Broiler	Jeolla	MW984629
Kr/D022/13	2013	Broiler	Jeolla	MW984630
Kr/D024/13	2013	Broiler	Jeolla	MW984631
Kr/D025/13	2013	Broiler	Jeolla	MW984632
Kr/Q022/13	2013	Broiler	Jeolla	MW984633
Kr/Q028/13	2013	Broiler	Jeolla	MW984634
Kr/Q030/13	2013	Broiler	Jeolla	MW984635
Kr/D068/14	2014	Native	Chungcheong	MW984636
Kr/D069/14	2014	Native	Chungcheong	MW984637
Kr/D062/16	2016	Native	Geyonggi	MW984638
Kr/D053/17	2017	Broiler	Gyeonsang	MW984639
Kr/R024/17	2017	Native	Chungcheong	MW984640
Kr/Q037/18	2018	Native	Gyeonsang	MW984641
Kr/D013/18	2018	Broiler	Jeolla	MW984642
Kr/D014/18	2018	Broiler	Geyonggi	MW984643
Kr/R030/18	2018	Broiler	Chungcheong	MW984644
Kr/R38A/18	2018	Native	Geyonggi	MW984645
Kr/D024/18	2018	Layer	Geyonggi	MW984646
Kr/D025/18	2018	Broiler	Jeolla	MW984647
Kr/Q041/18	2018	Broiler	Gangwon	MW984648
Kr/D030/18	2018	Broiler	Geyonggi	MW984649
Kr/D038/18	2018	Layer	Chungcheong	MW984650
Kr/R038/18	2018	Native	Geyonggi	MW984651
Kr/R109/18	2018	Native	Gyeonsang	MW984652
Kr/Q054/18	2018	Broiler	Jeolla	MW984653
Kr/Q057/18	2018	Broiler	Jeolla	MW984654
Kr/D048/18	2018	Broiler	Chungcheong	MW984655
Kr/D063/18	2018	Broiler	Geyonggi	MW984656
Kr/R084/18	2018	Broiler	Chungcheong	MW984657
Kr/Q063/18	2019	Broiler	Chungcheong	MW984658
Kr/D005/19	2019	Layer	Gyeonsang	MW984659
Kr/Q002/19	2019	Broiler	Chungcheong	MW984660
Kr/Q008/19	2019	Broiler	Gyeonsang	MW984661
Kr/D021/19	2019	Broiler	Jeolla	MW984662
Kr/R019/19	2019	Broiler	Gyeonsang	MW984663
Kr/R024/19	2019	Broiler	Gyeonsang	MW984664
Kr/R014/19	2019	Broiler	Gyeonsang	MW984665
Kr/Q010/19	2019	Broiler	Gyeonsang	MW984666
Kr/D030/19	2019	Broiler	Gyeonsang	MW984667
Kr/D031/19	2019	Layer	Geyonggi	MW984668
Kr/D034/19	2019	Broiler	Jeolla	MW984669
Kr/D038/19	2019	Broiler	Jeolla	MW984670
Kr/D039/19	2019	Broiler	Geyonggi	MW984671
Kr/D060/19	2019	Native	Chungcheong	MW984672
Kr/D068/19	2019	Layer	Geyonggi	MW984673
Kr/D072/19	2019	Broiler	Chungcheong	MW984674

**Table 2 viruses-13-01163-t002:** Tracheal ciliostasis scores in SPF chickens infected with IBV isolates.

Isolate	Recombinant Pattern	Ciliostasis Score ^A^			
		3 dpi	7 dpi	14 dpi	21 dpi
Kr/D030/18	rGI-19-II	0.0 ± 0.0 ^B^	4.0 ± 0.0	2.3 ± 0.6	0.9 ± 0.6
Kr/Q037/18	rGI-19-III	0.0 ± 0.1	2.1 ± 0.9	3.9 ± 0.2	2.6 ± 0.9
Kr/D025/18	GI-19	0.0 ± 0.0	3.7 ± 0.5	4.0 ± 0.0	0.2 ± 0.5
Control		0.0 ± 0.0	0.0 ± 0.0	0.0 ± 0.0	0.0 ± 0.0

^A^ Data are expressed as the mean score ± SD. ^B^ A ciliostasis score of 0, 1, 2, 4, and 4 corresponds to 100%, 75%, 50%, and 0% activity, respectively.

## Data Availability

The data presented in this study are available on request from the corresponding author.

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
