# Peer review of "Genetic and Pathogenic Characterization of QX(GI-19)-Recombinant Infectious Bronchitis Viruses in South Korea"

_viruses, 2021, doi:10.3390/v13061163_

Round 1

Reviewer 1 Report

The manuscript received presents results of genetic and pathogenic characterisation of QX-strains of IBDV identified in South Korea. The presented manuscript presents the results, interesting from both a practical and scientific point of view, of a comparative virulence study of two QX recombinants and the putative parental strain. The paper presented provides new information on the potential impact of recombination within the S gene on the tropism and virulence of IBV strains. The manuscript is well written; however, it requires some improvements.

The main comment concerns the IBV genotype nomenclature used. Recently, Valastro et al (2016) introduced new rules for the genotyping and nomenclature of IBVs, which are now widely used in the scientific literature. However, the authors use IBV genotype terms that may be unclear or confusing for the reader. Therefore, the nomenclature introduced by Valstro should be used, and for example QX strains belong to genotype GI-19, while strains marked in the article as K-1 to genotype GI-15.

The second comment concerns the lack of statistical analysis for results obtained for virus quantification in different tissues or swabs from the animal trial. The results presented are very interesting, but without statistical analysis it is difficult to draw correct conclusions.

There is also no information on the origin of the parental strain KM91, it seems that this strain also emerged by recombination event, is it known what the origin of the acquired insert was?

Minor following amendments are needed:

  • Lines 110: “Three groups were infected with 100 l of IBV (5 EID50 per 0.1 ml dose) via the oculo-nasal route” – Please correct the indicated unit of volume and use the readable notation of virus titre e.g. the exponent as superscript(the same in line 128)
  • Line 115: “ The remaining ten birds in each group were euthanatized (…)” According to the authors, the group consisted of 20 birds. A total of 11 chicks were euthanised (3 at 3 dpi and 4 each at 7 and 14dpi), so 9 rather than 10 birds remained. Please make corrections.
  • Figure 4. - In Figure C, the sign "*" was used but in the description of the figure there is no information what it means.
  • Lines 275-277: “The QX genotype also exhibits nephrotic tropism, causing gross lesions that are pale and swollen, with distention of the ureter and excess production of urates” Please rephrase this sentence as it is not clear. What is pale and swollen? I would also suggest using another term instead of "nephrotic" e.g. renal
  • Line 285: “in the trachea at 7 dpi an14 dpi (…)” Please correct the spelling mistake.
  • References are not numbered in order of appearance in the text, as stated in the requirements for authors

Author Response

Comments to the Author

The manuscript received presents results of genetic and pathogenic characterisation of QX-strains of IBDV identified in South Korea. The presented manuscript presents the results, interesting from both a practical and scientific point of view, of a comparative virulence study of two QX recombinants and the putative parental strain. The paper presented provides new information on the potential impact of recombination within the S gene on the tropism and virulence of IBV strains. The manuscript is well written; however, it requires some improvements.

Major comments

Point 1: IBV genotype nomenclature used. Recently, Valastro et al (2016) introduced new rules for the genotyping and nomenclature of IBVs, which are now widely used in the scientific literature. However, the authors use IBV genotype terms that may be unclear or confusing for the reader. Therefore, the nomenclature introduced by Valstro should be used, and for example QX strains belong to genotype GI-19, while strains marked in the article as K-1 to genotype GI-15.

Response 1: Thank you for pointing this out. All co-authors made an agreement that the new nomenclature of IBV by Valastro’s et al (2016) should be applied to the manuscript. However, according to Valastro’s classification, the nomenclature of the GI-19 subgroup (KM91-like and QX-like) was not determined yet. Therefore, we changed all genotype terms except GI-19 subgroup in the manuscript (Mass → GI-1, K-1-like → GI-15, and QX → GI-19).

Point 2: The lack of statistical analysis for results obtained for virus quantification in different tissues or swabs from the animal trial. The results presented are very interesting, but without statistical analysis it is difficult to draw correct conclusions

Response 2: Thank you for pointing this out. This part was added in Results and Figure 4-5 of the manuscript.

Point 3: There is also no information on the origin of the parental strain KM91, it seems that this strain also emerged by recombination event, is it known what the origin of the acquired insert was?

Response 3: Thank you for your question. KM91 is novel strain in South Korea and distinguished from other genotypes, but non-recombinant strain. Therefore, KM91 were used as putative parental strains to analysis recombinant in previous studies (reference 1-3).

Reference 1: Mo, M.L.; Hong, S.M.; Kwon, H.J.; Kim, I.H.; Song, C.S.; Kim, J.H. Genetic diversity of spike, 3a, 3b and E gene of infectious bronchitis viruses and emergence of new recombinants in Korea. Viruses 2013, 5, 550-567.

Reference 2: Lim, T.H.; Lee, H.J.; Lee, D.H.; Lee, Y.N.; Park, J.K.; Youn, H.N.; Kim, M.S.; Lee, J.B., Park, S.Y.; Choi, I.S.; Song, C.S. An emerging recombinant cluster of nephropathogenic strains of avian infectious bronchitis virus in Korea. Infect. Genet. Evol. 2011, 11, 678-685.

Reference 3: Song, C.S.; Lee, Y.N.; Kim, J.H.; Sung, H.W.; Lee, C.W.; Izumiya, Y.; Miyazawa, T.; Jang, H.K., Mikami, T. Epidemiological classification of infectious bronchitis virus isolated in Korea between 1986 and 1997. Avian Pathol. 1998, 27, 409-416.

Minor comments:

Regarding minor comments, we have now corrected the manuscript according to your suggestion:

Point 1: (Lines 110) “Three groups were infected with 100 l of IBV (5 EID50 per 0.1 ml dose) via the oculo-nasal route”? Please correct the indicated unit of volume and use the readable notation of virus titre e.g. the exponent as superscript (the same in line 128)

Response 1: According to the reviewer’s comment, we amended it in Lines 111 and 129

Point 2: (Line 115) “The remaining ten birds in each group were euthanatized (…)” According to the authors, the group consisted of 20 birds. A total of 11 chicks were euthanised (3 at 3 dpi and 4 each at 7 and 14dpi), so 9 rather than 10 birds remained. Please make corrections.

Response 2: According to the reviewer’s comment, we amended it in Lines 115-116 )

Point 3: (Figure 4) In Figure C, the sign "*" was used but in the description of the figure there is no information what it means.

Response 3: According to the reviewer’s comment, we amended it in Line 244

Point 4: (Lines 275-277) “The QX genotype also exhibits nephrotic tropism, causing gross lesions that are pale and swollen, with distention of the ureter and excess production of urates” Please rephrase this sentence as it is not clear. What is pale and swollen? I would also suggest using another term instead of "nephrotic" e.g. renal

Response 4: According to the reviewer’s comment, we amended it in Line 282-284

Point 5: (Line 285) “in the trachea at 7 dpi an14 dpi (…)” Please correct the spelling mistake.

Response 5: According to the reviewer’s comment, we amended it in Line 292

Point 6: References are not numbered in order of appearance in the text, as stated in the requirements for authors

Response 6: According to the reviewer’s comment, we amended it in manuscript

Reviewer 2 Report

In the manuscript of Youn et al the authors analysed S gene sequences of 56 IBV isolates collected from infected chickens in South Korea between 2003 and 2019. Ten isolates formed a new cluster, and recombination events in the S1 gene were identified. Finally, the authors compared pathogenicity of two recombinant viruses with a non-recombinant in 1-week-old SPF chickens.

Major comments:

  • There is no indication that methods of statistics were applied for analysis of the data presented in Fig. 4 and 5. Mark columns with statistically significant difference of the mean and show p value. In the text describing the results, clearly mention at what time points statistically significant differences were observed. In addition, statistical methods have to be mentioned in the Materials and Methods section.
  • Table 3 presents both text and pictures. This is unusual method of data presentation. You can leave text only in the table and not show pictures. Alternatively, you can create a new figure where you show these pictures and describe finding in the text of the manuscript.

Minor comments:

  • Correct typos in lines 47, 110 and 196.
  • Line 155 – replace “homology” with “identity”.
  • Fig 1 – you may explain in the text that “Mass” is a group of viruses related to the vaccine Massachusetts strain.  

Author Response

Comments to the Author

In the manuscript of Youn et al the authors analysed S gene sequences of 56 IBV isolates collected from infected chickens in South Korea between 2003 and 2019. Ten isolates formed a new cluster, and recombination events in the S1 gene were identified. Finally, the authors compared pathogenicity of two recombinant viruses with a non-recombinant in 1-week-old SPF chickens.

Major comments

Point 1: There is no indication that methods of statistics were applied for analysis of the data presented in Fig. 4 and 5. Mark columns with statistically significant difference of the mean and show p value. In the text describing the results, clearly mention at what time points statistically significant differences were observed. In addition, statistical methods have to be mentioned in the Materials and Methods section.

Response 1: Thank you for pointing this out. This part was added in Materials & Methods part, Result part, and Figure 4&5.

Point 2: Table 3 presents both text and pictures. This is unusual method of data presentation. You can leave text only in the table and not show pictures. Alternatively, you can create a new figure where you show these pictures and describe finding in the text of the manuscript.

Response 2: Thank you for your suggestion. According to the reviewer’s comment, we created a new figure (Figure 4) and described in the manuscript.

Minor comments:

Regarding minor comments, we have now corrected the manuscript according to your suggestion:

Point 1: Correct typos in lines 47, 110 and 196.

Response 1: According to the reviewer’s comment, we amended it in Lines 48, 111, and 201

Point 2: (Line 155) replace “homology” with “identity”.

Response 2: According to the reviewer’s comment, we amended it in Line 161

Point 3: (Fig 1) you may explain in the text that “Mass” is a group of viruses related to the vaccine Massachusetts strain

Response 3: According to the reviewer’s comment, we amended it in Line 141

Round 2

Reviewer 1 Report

I don't have any additional comments.

Reviewer 2 Report

My recommendations were accepted.